# Drimane Sesquiterpene Alcohols with Activity against *Candida* Yeast Obtained by Biotransformation with *Cladosporium antarcticum*

**DOI:** 10.3390/ijms232112995

**Published:** 2022-10-27

**Authors:** Nicole Cortez, Víctor Marín, Verónica A. Jiménez, Víctor Silva, Oscar Leyton, Jaime R. Cabrera-Pardo, Bernd Schmidt, Matthias Heydenreich, Viviana Burgos, Paola Duran, Cristian Paz

**Affiliations:** 1Laboratory of Natural Products & Drug Discovery, Department of Basic Science, Center CEBIM, Universidad de La Frontera, Av. Francisco Salazar 01145, Temuco 4811230, Chile; 2Departamento de Ciencias Químicas, Facultad de Ciencias Exactas, Universidad Andres Bello, Sede Concepción, Autopista Concepción-Talcahuano 7100, Talcahuano 4260000, Chile; 3Tecnología Médica, Facultad de Salud, Universidad Santo Tomás, Temuco 4780000, Chile; 4Laboratorio de Química Aplicada y Sustentable (LabQAS), Departamento de Química, Facultad de Ciencias, Universidad del Bío-Bío, Concepcion 4081112, Chile; 5Institut für Chemie, Universität Potsdam, Karl-Liebknecht-Str. 24-25, D-14476 Potsdam, Germany; 6Departamento de Ciencias Básicas, Universidad Santo Tomás, Temuco 4780000, Chile; 7Biocontrol Research Laboratory, Universidad de La Frontera, Temuco 4811230, Chile

**Keywords:** *Drimys winteri*, *Cladosporium antarcticum*, drimendiol, epidrimendiol, biotransformation, *Candida* yeast, lanosterol 14α-demethylase, biocontrol

## Abstract

Fungal biotransformation is an attractive synthetic strategy to produce highly specific compounds with chemical functionality in regions of the carbon skeleton that are not easily activated by conventional organic chemistry methods. In this work, *Cladosporium antarcticum* isolated from sediments of Glacier Collins in Antarctica was used to obtain novel drimane sesquiterpenoids alcohols with activity against *Candida* yeast from drimendiol and epidrimendiol. These compounds were produced by the high-yield reduction of polygodial and isotadeonal with NaBH_4_ in methanol. *Cladosporium antarcticum* produced two major products from drimendiol, identified as 9α-hydroxydrimendiol (**1**, 41.4 mg, 19.4% yield) and 3β-hydroxydrimendiol (**2**, 74.8 mg, 35% yield), whereas the biotransformation of epidrimendiol yielded only one product, 9β-hydroxyepidrimendiol (**3**, 86.6 mg, 41.6% yield). The products were purified by column chromatography and their structure elucidated by NMR and MS. The antifungal activity of compounds **1**–**3** was analyzed against *Candida albicans*, *C. krusei* and *C. parapsilosis*, showing that compound **2** has a MIC lower than 15 µg/mL against the three-pathogenic yeast. In silico studies suggest that a possible mechanism of action for the novel compounds is the inhibition of the enzyme lanosterol 14α-demethylase, affecting the ergosterol synthesis.

## 1. Introduction

Non-albicans *Candida* species have become a major sanitary threat owing to their growing prevalence among immunocompromised and intensive care unit patients. Current treatments for infections caused by these strains face serious challenges arising from the emergence of antifungal resistance to conventional drugs. For example, *Candida auris* has shown resistance to fluconazole (93%), amphotericin B (7%), echinocandin (42%) and 4% to all antifungal classes [1]. In this scenario, there is a growing interest in developing novel compounds capable of overcoming the drawbacks of conventional antifungal agents to tackle needs related to fungal infections worldwide.

Drimane sesquiterpenoids are a family of natural and semi-synthetic compounds with promising applications to treat conditions that present intrinsic or secondary resistance to conventional antifungals. Among them, the α,β-unsaturated 1,4-dialdehyde polygodial presents remarkable potential owing to its potent activity against a broad spectrum of fungi, including *Botrytis cinerea* [2], *Gaeumannomyces graminis* var. tritici [3] and parasitic yeast such as *Candida albicans* with an MIC of 3.13 µg/mL [4]. The activity of polygodial against *Candida* yeast has been associated with the lipophilic nature of the drimane skeleton, and the capability to form pyrrole derivatives with nucleophiles as amines, which can disrupt the lipid–protein interface as a non-ionic surface-active agent [5,6,7,8]. Polygodial is also an agonist of the receptor hTRPA1 [9] and inhibits voltage-gated sodium channels involved in pain sensation, Nav1.7 and Nav1.8 with an MIC of 16 ± 8 and 57 ± 7 µM, respectively [10]. Additionally, polygodial is toxic for parasites such as *Trypanosoma cruzi* [11] and displays insecticidal and deterrent properties [12]. Despite its potent activity, the clinical use of polygodial is hampered by its pungency and irritating properties against eyes, sensible skin or injuries. These drawbacks raise the need and opportunity for exploring the synthesis of novel drimane sesquiterpene derivatives with improved antifungal activity and reduced undesired effects.

This work focuses on the study of drimane sesquiterpene alcohols obtained by the reduction of the dialdehydes polygodial and isotadeonal with NaBH_4_ in methanol. The parent aldehydes differ in the configuration of C9 and lead to reduced alcohols with inverted stereochemistry (*R* from polygodial and *S* from isotadeonal). The obtained alcohols were further functionalized to obtain a series of drimane derivatives by taking advantage of green methods of synthesis, such as biotransformations mediated by enzymes, bacteria or fungi. To this aim, we isolated the fungus *Cladosporium antarcticum* from Glacier Collins sediments in Antarctica and used it to produce three compounds identified by 1D- and 2D-NMR. The activity of the biotransformed products was evaluated against *Candida albicans*, *Candida krusei* and *Candida parapsilosis*. Additionally, in silico modeling through molecular dynamic simulations was carried out to examine the interaction of the biotransformation products with the enzyme lanosterol-14α demethylase, which is a target for drimane sesquiterpenoids activity [13]. The tested compounds showed activities around 15 µg/mL, which are promising to develop more potent compounds from these chemical scaffolds for *Candida* yeast treatment.

## 2. Results

### 2.1. Identification of Yeast Samples by MALDI-TOF-MS

The yeasts *Candida albicans*, *Candida krusei* and *Candida parapsilosis* were acquired from Universidad Mayor (Chile) and subjected to proteomic analysis for yeast identification. The yeasts were cultured for 48 h in the dark at 37 °C, and fresh colonies were analyzed by MALDI-TOF MS after a careful equipment calibration protocol with the protein calibration standard I (insulin, ubiquitin, cytochrome C and myoglobin). The protein fingerprint of each yeast was compared with a library of 1301 spectra of fungi in the range of *m*/*z* 3000–15,000. The identification results are presented as logarithmic scores ranging from 0 to 3.0. Scores above 1.7 were used for genus identification, whereas values higher than 2.0 were used for species assignment (Table 1).

### 2.2. Isolation of Cladosporium antarcticum

The fungus isolated from Collins Glacier (Figure 1A) revealed the presence of tubular structures on conidiophore and conidium according to the scanning electron micrographs (SEM) of spores (Figure 1B). The identification and phylogenetic affiliation of the strain were carried out based on the sequencing of the ribosomal internal transcribed spacer 2 (ITS2) region and by sequence comparison with other *Cladosporium antarcticum* strains present in the GenBank database. According to these findings, the isolated strain corresponds to *Cladosporium antarcticum* (Accession N° OP272990) (Figure 2).

### 2.3. Biotransformation of Drimendiol and Epidrimendiol

Drimendiol and epidrimendiol were obtained by the straightforward reduction of the sesquiterpene dialdehydes polygodial and isotadeonal with two equivalents of NaBH_4_ in MeOH at room temperature. The corresponding reaction yields were 98% and 95% for drimendiol and epidrimendiol, respectively. Then, 200 mg of the obtained drimane sesquiterpene diols was biotransformed by *Cladosporium antarcticum* for five days, at 18 °C, under constant stirring (1200 rpm). After the products’ purification by column chromatography, it was determined that drimendiol produced two major products, identified as 9α-hydroxydrimendiol (**1**, 41.4 mg, 19.4% yield) and 3β-hydroxydrimendiol (**2**, 74.8 mg, 35% yield), as detailed in Figure 1. On the other hand, epidrimendiol led to a single major product, identified as 9β-hydroxyepidrimendiol (**3**, 86.6 mg, 41.6% yield), as displayed in Figure 2. The biotransformation products **1, 2** and **3** were subjected to an exhaustive characterization by 1D- and 2D-NMR spectroscopy. The results obtained from ^1^H-NMR and ^13^C-NMR data are summarized in Table 2 and Table 3.

### 2.4. Anti-Candida Activity

The antifungal activities of the biotransformation products **1**–**3** were evaluated from minimal inhibitory concentration (MIC) measurements against *Candida parapsilopsis*, *Candida krusei* and *Candida albicans* using the broth microdilution method. The activities of the parent compounds drimendiol and epidrimendiol were also measured for comparative purposes. All experiments were carried out in three independent experiments in triplicate. Results were compared with the activity of fluconazole as a reference antifungal compound and are summarized in Table 4.

### 2.5. In Silico Studies

Molecular Dynamic (MD) simulations were carried out to examine the potential of compounds **1**–**3** to bind to the active site of the enzyme lanosterol 14α-demethylase as a potential mechanism for their antifungal activity. The initial coordinates for the ligand–enzyme complexes were obtained from blind docking calculations against the enzyme to avoid biases. For all ligands, the highest-ranked structure was found to place the ligand within the catalytic pocket nearby the HEM group (Figure 3A). The time evolution of the corresponding binding modes was examined from unrestrained 150 ns MD trajectories. Ligand root mean square deviation (RMSD) analysis in protein-aligned trajectories revealed that all compounds remain bonded to the catalytic pocket with mean RMSD values between 1.2–3.3 Å. The largest ligand mobility was found for compound **1**; nevertheless, it does not undergo a significant displacement from the binding pose predicted from molecular docking calculations (Figure 3B). Thereby, 150 ns MD simulations confirm that all ligands are capable of forming stable complexes with lanosterol 14α-demethylase and that preferred interactions occur at the catalytic pocket of the enzyme.

The nature of the intermolecular interactions responsible for ligand–enzyme binding was examined using the Protein–Ligand Interaction Profiler web tool [14] on 150 frames retrieved from the equilibrated MD trajectories. (Figure 4). Results indicate that the binding of a ligand is mostly mediated by hydrophobic interactions between the drimane moiety and apolar residues of the binding site. Hydrophobic interactions are mostly driven by Tyr126, Phe134 and Phe236 residues, which participate in the binding of all studied ligands. On the other hand, the residues Tyr140, Gly314 (backbone), Thr318 and Phe134 (backbone) participate in hydrogen bonds. The diversity of interactions found within the set of complexes under study suggests that slight chemical and stereochemical differences between ligands affect their binding modes to the enzyme. This could be related to differential inhibitory capacities.

Ligand–enzyme binding free energy calculations were carried out using the MM/GBSA approach to estimate the strength of the corresponding complexes (Figure 5). According to these calculations, all ligands are capable of binding to lanosterol 14α-demethylase with similar binding free energy. Nevertheless, compounds **2** and **3** show significantly lower mean values than their parent compounds drimendiol and epidrimendiol (*p* < 0.05). Decomposition of the gas-phase binding free energy in van der Waals and electrostatic terms suggests that the lower binding free energy values for compounds **2** and **3** arise from larger electrostatic terms, which in turn arise from the polarity increase after the biotransformation process.

## 3. Discussion

Pathogen drug resistance is a major global concern for public health worldwide. Valuable opportunities to tackle this challenge arise from the development of novel bioactive compounds capable of addressing the needs of current and future antifungal therapies. As a contribution to this research field, herein, we explored the synthesis of drimane sesquiterpene alcohols by chemical and biochemical methods, looking for bioactive compounds that can control parasitic *Candida* yeast with fewer side effects than polygodial. Polygodial is a drimane dialdehyde isolated from *Drimys winteri* with potent activity against *Candida albicans* yeast but unsuitable for therapeutic use owing to its pungency and irritant properties. Our approach in this regard started with the exploration of natural compounds with aldehyde moieties, which have shown promising bioactive properties. For example, the aldehyde ugandensidial showed activity against *Klebsiella pneumoniae*, *Moraxella catarrhalis, Pseudomonas aeruginosa* and *Staphylococcus aureus*, with MIC values of 0.130, 0.104, 0.078 and 0.130 mg/mL, while the dialdehyde warburganal showed activity of 0.130, 0.208, 0.104, 0.156 mg/mL against the same microorganism [15]. Moreover, warburganal is more potent against *C. albicans* and *C. glabrata*, with a MICs of 4.5 and 50 μg/mL, respectively [16]. Polygodial also inhibits the voltage-gated Na^+^ channels Na_V_1.7 and Na_V_1.8 involved in anesthesia sensation [10]. Despite these promising properties, the possible therapeutic use of aldehydes is controversial due to the promiscuous reactivity of the formyl moiety against nucleophilic groups, such as in lysine, to form pyrrole groups in biochemical environments. This covalent mechanism has been described for polygodial, which modulates TRPA1, an analgesic target protein, via covalent bonding with lysine residues [17]. Thereby, in this work, we focused on the isolation of the drimane sesquiterpenoids dialdehydes polygodial and isotadeonal on a gram scale, and their subsequent reduction to drimane sesquiterpenoids diols via a fast, mild and efficient reaction with NaBH_4_ in methanol. The obtained alcohols, drimendiol and epidrimendiol, showed antifungal activity with MIC values from 12 to 50 μg/mL. These compounds were further biotransformed by *Cladosporium antarcticum* as a green strategy to develop novel bioactive species that are not easily obtained by traditional organic chemical methods. The biotransformation of drimendiol produced two compounds, identified as 9α-hydroxydrimendiol (**1**) and 3β-hydroxydrimendiol (**2**). When the substrate was epidrimendiol, the fungus produced only one product, 9β-hydroxyepidrimendiol (**3**). Compound **1** was previously reported by Asakawa et al. in 2018 [18] as the product of the biotransformation of drimendiol by *A. niger* in 3 days. In that report, the reaction yield and products depended largely on the incubation time. In our process, *Cladosporium antarcticum* seemed to be selective in the production of compounds **1** and **2** during the 5 days of incubation. The activity of all biotransformation products is similar, but compound **2** seems to be slightly more active than others against *C. parapsilosis*, *C. krusei* and *C. albicans*, showing MIC values of 12.5, 15.0 and 15.0 μg/mL, respectively. In the case of *C. parapsilosis*, the activities of compounds **1**–**3** are similar to the reference compound fluconazole, which are auspicious and encourage continued exploration of these molecular scaffolds as potential antifungal agents. Within the evaluated species, compound **2** showed the lowest MIC values against the three tested strains. In silico studies were carried out to evaluate the suitability of compounds **1**–**3** to bind to the catalytic pocket of the enzyme lanosterol 14α-demethylase, which is a proposed target for drimane sesquiterpenes with antifungal properties. Our findings suggest that all compounds are capable of binding effectively with the enzyme, which is in line with the observed activity against *Candida* yeasts for compounds **1**–**3** and their parent species drimendiol and epidrimendiol. Nevertheless, compounds **2** and **3** showed lower binding free energies than their precursors, which is concordant with the in vitro assays. The strengthening of ligand–enzyme interactions arises from the larger electrostatic terms in the calculated gas-phase binding free energies for compounds **2** and **3** compared with the parent species. It remains in any case to understand the toxicity of these compounds and additional mechanisms of action in the yeast, such as hyphae transition and biofilm formation, which will be assessed in further studies.

## 4. Material and Methods

### 4.1. Candida Yeast

*Candida albicans*, *Candida krusei* and *Candida parapsilopsis* were obtained from Culture Collection at Mayor University in Santiago, Chile [19]. The yeasts were isolated from Chilean patients with candidemia. Each strain was transferred from a cryo-tube to a Petri dish with Sabouraud dextrose agar and incubated at 35 ± 2 °C for 48 h. The identification of each strain was carried out by morphological, biochemical and proteome fingerprint evaluation. In brief, *Candida* species were characterized by germ tube analysis, morphological evaluation on cornmeal-tween 80 agar and carbohydrate assimilation patterns [20].

### 4.2. Proteome Fingerprinting MALDI-TOF MS

Fresh colony yeast was applied directly to a sample plate of the equipment, the samples were coated with a saturated solution of α-cyano acid 4-hydroxy cinnamic diluted in 50% ACN with 2.5% TFA. The Mass spectra were recorded using a MALDI-TOF MS Autoflex Speed (Bruker Daltonics, Bremen, Germany) equipped with an intelligent beam laser source (334 nm). The analyses were analyzed in linear mode with positive polarity, acceleration voltage of 20 kV and delay extraction of 220 ns. Each spectrum was collected as an average of 1200 laser shots with enough energy to produce good spectra without saturation in the range of *m*/*z* from 2000 to 20,000. Before analysis, the equipment was externally calibrated with the protein calibration standard I (Bruker Daltonics, Bremen, Germany, insulin, ubiquitin, cytochrome C and myoglobin) with FlexControl 1.4 software (Bruker Daltonics, Bremen, Germany). The analyses were analyzed with the MALDI Biotyper Compass 4.1 software (Bruker Daltonics, Bremen, Germany) in the range of *m*/*z* 3000–15,000 compared with a library of 1301 spectra of fungal identifications.

### 4.3. Isolation of Cladosporium antarcticum from Antarctic Sediments

Sediments from the base of Collins Glacier were collected in February 2019 (430 g, S 62°09′51″ W 58°51′00″) during the Antarctic campaign ECA-57 organized by the Instituto Antartico Chileno (INACH). The soil sample was transported into a cooler at 4 °C to the Laboratory of Natural Products & Drug Discovery of the University of La Frontera in Temuco, Chile. Fungi were isolated according to the procedures of Gonçalves et al. 2015 [21]. To resume, 1 g of sediment was added to 9 mL of sterile saline solution (0.85% NaCl) and vortexed, then 1 mL of the supernatant was diluted 10 and 100 times with saline solution. A total of 100 µL of each dissolution was spread onto a Petri dish with agar YM media (0.3% yeast extract, 0.3% malt extract, 0.5% peptone, 2% glucose, 2% agar and pH 6.2 ± 2) in triplicate. Plates were supplemented with chloramphenicol (100 μg mL^−1^) to prevent bacterial growth. The plates were incubated at 15 for 30 days, then cultures were isolated and grown again to obtain pure cultures visualized by scanning electron microscopy (VP-SEM) with an energy dispersive X-ray spectrometer detector (EDX, Hitachi, Japan).

Filamentous fungi were purified from 1:100 dilutions and identified by sequencing of the ribosomal internal transcribed spacer 2 (ITS2) region. ITS2 was amplified by touchdown polymerase chain reaction (PCR) with the primer set fITS9 (5′-GAACGCAGCRAAIIGYG-3′) and ITS4 (5-′TCCTCCGCTTATTGATATGC-3′) as described by Duran et al. (2019) [22], using the following conditions: initial denaturation at 95 °C for 3 min, followed by 25 cycles each at 95 °C for 30 s, an annealing step with a 0.5 °C decrease each cycle from 65 to 52.5 °C and extension at 72 °C for 30 s. Twenty-five additional cycles were carried out with denaturation at 95 °C for 30 s and a 55 °C annealing step and primer extension at 72 °C for 30 s, with a final extension step of 7 min at 72 °C. The PCR products were purified and sequenced by Smart C- BIOREN Universidad de la Frontera.

### 4.4. General Chemical Information

Analytical thin-layer chromatography (TLC) was carried out on Merck Silica Gel 60F254 sheets (Darmstadt, Germany). It was employed for monitoring the purification process and the reaction progress of new compounds, using a mixture of *n*-hexane: EtOAc = 1:1 and molybdophosphoric acid for visualization. Preparative chromatography was performed using Merck silica gel 60 (CC) and Sephadex LH-20 (25−100 μm; Aldrich, Santiago, Chile). Solvents and fractions were concentrated in vacuo using a Büchi R100 rotavap. Solvents used in this study were distilled before use and dried over appropriate drying agents.

### 4.5. Reduction of Polygodial and Isotadeonal with Sodium Borohydride

To a solution of polygodial or isotadeonal (234 g/mol, 200 mg, 0.85 mmol) in methanol (30 mL) 2 equivalents of sodium borohydride (NaBH_4_, 37.8 g/mol, 65 mg) were added dropwise, then the reaction mixture was stirred at room temperature for 30 min. After the reaction was completed, followed by TLC, HCl 0.1 M was added until pH 2. The reaction was extracted with EtOAc (100 mL, three times) and concentrated under vacuum. The crude extract was purified by CC with hexane/EtOAc (1:1 *^v^*/*_v_*).

### 4.6. Biotransformation

To a five-day culture (300 mL, YM media, 20 °C, under stirring) of *Cladosporium antarcticum* a solution of drimendiol or epidrimendiol (200 mg, 238 g/mol, 0.84 mmol) was added dropwise and dissolved in two mL of acetone, then the fungal culture was kept under growing conditions. Each day, the biotransformation process was monitored, removing five mL of the liquid culture (under sterile conditions), extracting the organic compounds with EtOAc and evaluating the starting material and new products by TLC. After five days, the biotransformation was stopped, the liquid media was extracted three times with 300 mL of EtOAc and the organic layer was pooled and dried with anhydrous MgSO_4 (s)_, then the solvent was removed under vacuum, obtaining a crude extract, which was purified by silica gel column chromatography with hexane/EtOAc (1:1 *^v^*/*_v_*). The products were dried and kept at −20 °C.

### 4.7. Structural Analysis of Drimane Sesquiterpenoids

The structures of compounds produced by the biotransformation of drimendiol (**3** and **4**) and epidrimendiol (**7**) were elucidated by 1D- and 2D-NMR. The ^1^H- and ^13^C-NMR spectra were recorded in CDCl_3_ solution in 5 mm tubes at RT on a Bruker Avance III 400 MHz spectrometer (Bruker Biospin GmbH, Rheinstetten, Germany), with the deuterium signal of the solvent as the lock and TMS (for ^1^H) or the solvent (for ^13^C) as internal standard. All spectra (^1^H, ^13^C, gs-H,H−COSY, edited HSQC and gs-HMBC) were acquired and processed with the standard Bruker software (TopSpin 4.x, Bruker, Germany).

### 4.8. Antifungal Assay

The in vitro antifungal assay was performed in triplicate by a broth microdilution method following Clinical Laboratory Standard Institute (CLSI) recommendations according to the document M27A. In summary, the samples were dissolved in dimethyl sulfoxide (DMSO) stock solution at 1200 μg/mL and then diluted with sterile water in a work solution at 120 μg/mL. Compounds were evaluated in 96-well microplates between 60 to 6.25 μg/mL by serial dilutions in culture media (RPMI, 100 µL). The inoculum (100 µL) was adjusted to yield a cell concentration of 2.5–5.0 × 10^5^ UFC/mL. A microorganism growth control and one non-inoculated well were included to ensure medium sterility. The plates were incubated at 37 °C for 48 h in a humid chamber. Then, the growth in the positive control column and its absence in the negative control were visually verified. The MIC values were determined as the lowest concentrations of each compound capable of inhibiting microorganism growth by visual inspection compared with growth control. MIC values were determined by triplicate, and each assay was performed three times using different starting yeast. Fluconazole was used as a positive control at concentrations of 50, 25, 12.5, 6.25, 3.13, 1.56 and 0.78 µg/mL.

### 4.9. Molecular Simulation

Molecular dynamics simulations were carried out following the experimental protocol described in our previous works. Briefly, the coordinates of the enzyme lanosterol 14α-demethylase were taken from the crystallographic model 4LXJ from the PDB database [23]. Protein discontinuities were fixed as per the UNIPROT code P10614, and protonation states were set to pH 6.5 using the H++ web server [24]. The coordinates for the ligands were obtained from energy minimization with the MMFF94 force field using 3500 steepest descent steps and a convergence criterion of 10^−7^ kcal mol^−1^Å^−1^ in the Avogadro software [25]. Blind ligand–protein docking calculations were carried out using the CB-Dock web server [26,27]. The highest-ranked binding modes for each ligand were used as input structures for fully atomistic molecular dynamics simulations with the AMBER 20 software [28]. GAFF-consistent force field parameters for the ligands were obtained using AM1-BCC charges in antechamber. HEME parameters were retrieved from the AMBER parameter database available at http://amber.manchester.ac.uk/. The force field ff19SB was used to model the protein structure. Systems were solvated in an octahedral box of OPC waters extended 10 Å from the outermost atom of the complex. Systems were neutralized with proper counterions to keep charge neutrality. The pmemd.CUDA program was used to conduct MD simulations using the following protocol: (a) 1500 steepest descent minimization steps followed by 3500 conjugate gradient minimization steps for water molecules relaxation, (b) 1500 steepest descent minimization steps followed by 6500 conjugate gradient minimization steps for the entire system, (c) 500 ps of progressive NVT heating from 0 to 300 K (d) 500 ps of NVT equilibrium at 300 K with restrains applied to the protein backbone, (e) 20 ns of NVT equilibrium at 300 K and finally (f) 150 ns of unrestrained NPT production dynamics with a 2 fs time step at 300 K and 1 bar from which production data were collected. During MD simulations, the cutoff for non-bonded terms was 10 Å, long-range electrostatics were treated using the particle-mesh Ewald approach, and the SHAKE algorithm was employed to constrain all bonds involving hydrogen. Trajectory analysis was carried out using CPPTRAJ software. The MMPBSA.py module was used to estimate the strength of protein–ligand interactions under a single trajectory approach [29]. GB calculations were carried out using the modified GB model (igb = 5) with mbondi2 and α, β and γ values of 1.0, 0.8 and 4.85, respectively. Dielectric constants for the solvent and the protein were set to 80 and 1, respectively. A salt concentration of 0.15 mol L^−1^ was considered in this process to mimic physiological conditions for binding free energy estimates. The entropic term was not included in our calculations due to the size of the systems under study.

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
