# Peer review of "Drimane Sesquiterpene Alcohols with Activity against Candida Yeast Obtained by Biotransformation with Cladosporium antarcticum"

_ijms, 2022, doi:10.3390/ijms232112995_

Round 1
Reviewer 1 Report
The manuscript concerns the proces of obtaining new antifungal derivatives of the sesquiterpene aldehydes, polygodial and isotadeonal, which underwent reduction to yield drimane sesquiterpene alcohols. These alcohols were then subjected to biotransformation with Cladosporium antarcticum isolated from the Collins glacier sediments in the Antarctic. The formed compounds exhibited activity at ca. 15 μg/ml, which was considered promising in medication of the candidosis. The manuscript is a combination of experimental and computational approaches. It generally deserves to be published in the International Journal of Molecular Sciences with the following minor issues taken into account:
1. Table 2 is not provided in the manuscript.
2. The authors provide that the biotransformation process was monitored. How was the transformation of the substrates and the appearance of the products changing during the processes? Why was the transformation stopped on day 5? Perhaps it would be possible to obtain products with greater efficiency (yield), increasing the time of biotransformation.
3. Lines 343-344: the force field name is rather MMFF94; the convergence criterion lacks units (kcal/mol per Angstrom or bohr?). The time step of the molecular dynamics is also not given.
4. It might be (optionally) worthwhile to include the RMSD of the proteins along the trajectory. This could potentially show if ligand binding affects the structural stability of the protein. This would also require calculations on an empty protein (without any ligand).
Author Response
The authors sincerely thank the reviewers for their valuable comments on our work. In the revised version of the manuscript, we carefully addressed each comment and provided a detailed response in this document. Additionally, careful English language and grammar revisions were conducted to correct the minor mistakes pointed out by the referees after the revision process.
Reviewer 1
The manuscript concerns the process of obtaining new antifungal derivatives of the sesquiterpene aldehydes, polygodial and isotadeonal, which underwent reduction to yield drimane sesquiterpene alcohols. These alcohols were then subjected to biotransformation with Cladosporium antarcticum isolated from the Collins glacier sediments in the Antarctic. The formed compounds exhibited activity at ca. 15 μg/ml, which was considered promising in medication of the candidosis. The manuscript is a combination of experimental and computational approaches. It generally deserves to be published in the International Journal of Molecular Sciences with the following minor issues taken into account:
- Table 2 is not provided in the manuscript.
Response. The authors sincerely apologize for this mistake. In the revised version of the manuscript, the numbering was corrected.
- The authors provide that the biotransformation process was monitored. How was the transformation of the substrates and the appearance of the products changing during the processes? Why was the transformation stopped on day 5? Perhaps it would be possible to obtain products with greater efficiency (yield), increasing the time of biotransformation.
Response. We started this research using different fungi including A. niger. This system produced different products each day. On the other hand, C. antarcticum produced specific products with both substrates. Drimendiol led to two products (Scheme 1), whereas epidrimendiol only yielded one product. The produced compounds did not change after 5 days. For this reason, we decided to focus on the isolation and structural characterization of the products. According to our experience in the area, larger biotransformation times (> 5 days) could result in lower yields.

Reviewer 2 Report
The article is interesting. Authors present drimane sesquiterpene alcohols with activity against Candida. I suggest some corrections:
1. In the Methods, authors described that studied MICm what is proper. However, in the Abstract and Discussion IC50 values are presented. MIC is not IC50. Moreover, I checked that e.g. in reference [15] MIC was studied, but in the Discussion authors wrote that IC50. Please correct and change all IC50 names into the MIC. For bacteria and fungi, the MIC is precisely tested. IC50 is only given for viruses.
2. In "4.9. Molecular simulation", the authors studied the 4LXJ protein. Unfortunately, it is a protein of Saccharomyces cerevisiae, not Candida. It should be studied lanosterol 14α demethylase of Candida.
3. Why was studied only lanosterol 14α demethylase? Why has not been studied molecular simulation against other Candida enzymes that may be involved in antifungal activity?
Author Response
The authors sincerely thank the reviewers for their valuable comments on our work. In the revised version of the manuscript, we carefully addressed each comment and provided a detailed response in this document. Additionally, careful English language and grammar revisions were conducted to correct the minor mistakes pointed out by the referees after the revision process.
